# Clustering at the Disposal of Industry 4.0: Automatic Extraction of Plant Behaviors [note 1]

**DOI:** 10.3390/s22082939

**Published:** 2022-04-12

**Authors:** Dylan Molinié, Kurosh Madani, Véronique Amarger

**Affiliations:** LISSI Laboratory EA 3956, Sénart-FB Institute of Technology, Campus of Sénart, University of Paris-Est Créteil, 36-37 Rue Georges Charpak, F-77567 Lieusaint, France; amarger@u-pec.fr

**Keywords:** Industry 4.0, automatic behavior identification, automatic characterization, machine learning, data mining, clustering, quantification metrics

## Abstract

For two centuries, the industrial sector has never stopped evolving. Since the dawn of the Fourth Industrial Revolution, commonly known as Industry 4.0, deep and accurate understandings of systems have become essential for real-time monitoring, prediction, and maintenance. In this paper, we propose a machine learning and data-driven methodology, based on data mining and clustering, for automatic identification and characterization of the different ways unknown systems can behave. It relies on the statistical property that a regular demeanor should be represented by many data with very close features; therefore, the most compact groups should be the regular behaviors. Based on the clusters, on the quantification of their intrinsic properties (size, span, density, neighborhood) and on the dynamic comparisons among each other, this methodology gave us some insight into the system’s demeanor, which can be valuable for the next steps of modeling and prediction stages. Applied to real Industry 4.0 data, this approach allowed us to extract some typical, real behaviors of the plant, while assuming no previous knowledge about the data. This methodology seems very promising, even though it is still in its infancy and that additional works will further develop it.

## 1. Introduction

Deeper and more accurate understandings of systems are becoming essential in their control–command chain, particularly as systems are getting more and more complex [1]. This knowledge is highly valuable to keep track of a system’s evolution, to keep the process as smooth as possible and to prevent any unwanted events from occurring [2].

In Industry 4.0, the potential problems are numerous: bad communication between the processes [3,4], local disturbances propagated to the whole system [5], process drift leading to a drop in product quality, time delay, or even a more severe failure [6,7], etc.

This is not new in the industrial sector that a complex system can go the wrong way if it is left alone for too long; it is for this reason that the control–command theory has been developed for several decades [8,9]. Monitoring and controlling a system is not an easy task, since that requires defining upstream its “right” and “wrong” ways to behave; the most common way to do so is to study the system and to create its finite-state automaton, so as to follow its temporal evolution and to control it thereafter if needed [10,11,12,13,14].

Such automaton relies on the states that a system can achieve [15]; there are two categories of states: “regular” and “abnormal”. The former involves those that a system was designed to enter in, while the latter might involve temporary disturbances or failures [16]. For example, let us consider an aircraft’s aileron: it is designed to rotate and take any angle within a given interval; any value inside this interval is a regular state of the aileron, whereas any outside value represents an abnormal one. This example also illustrates the importance of knowing the regular states of a system, since if the aileron takes a value outside of its “normal” range, it can damage the trailing edge and the whole wing.

In a real dynamic system, the regular states are not as obvious, since many parameters and processes are connected, which greatly hardens the manual modeling of how each parameter impacts the others [17]. Even though the modeling of a system is traditionally manually crafted through expert system approaches, the ever-increasing complexities of industrial systems often make these methods too long, expensive, and difficult to complete; a solution that has been gaining ground for several years [18,19] involves using data mining so as to automatically extract some form of knowledge from raw data, such as the regular states of a system [20]. Once these states are identified, this information (already interesting by itself because it increases the understanding one has of the system and on how the processes are connected to one another) can also be used to further model the system, using either a multi-model [21,22,23,24,25] or a multi-agent [26,27,28,29,30] approach.

Note that, thus far, we have discussed a “regular state”, but if this state lasts for quite a long time (such as a system’s steady or shutdown states), it would sound more reasonable to call them “behaviors” rather than simply “states”, since they are representative of the real ways in which a system actually behaves. The questions on how to identify and characterize the behaviors of a system are still opened; these are generally the first steps to take prior to proceeding with a multi-model-based approach [21,31].

Interestingly, many works are based on the behaviors of a system, but few propose a real solution to the behavior identification problem; they mostly focus on how to use them once obtained, and are satisfied with common tools, such as artificial neural networks (ANNs) or “simple” mimicking of experts. Reference [32] proposed using behavior identification in the domain of malware detection and classification by monitoring the resources usage of a computer. In the context of an autonomous vehicle, references [33,34] based their behavior identification on the controlled study of the actions of car drivers, whilst [35,36] used a classical ANN. These works, although interesting, do not propose real ways to identify the behaviors of a system when having access to its raw data only.

Nonetheless, several works have directly addressed the problem of identifying the behaviors of a system so as to model it using finite-state automatons or multi-models. Such models serve two purposes: (1) identifying and preventing the anomalies [37,38,39,40,41,42]; (2) predicting the evolution of the system, either for real-time monitoring or for simulation [21,42,43,44,45]. Whether it is for fault detection and diagnosis or for prediction, the first step is to detect, identify, and characterize the different behaviors of the system.

In the domain of fault detection, references [43,44] proposed HyBUTLA, an automated building of the automaton of a hybrid plant by learning from its observations (a hybrid plant/system comprises both analog and continuous processes, and digital and discretized sensors, regulators, and control systems; thus, a real hybrid model should be a mixture of both analog and digital parts). Beginning with the different system components, and their associated events, it builds an automaton, refines it in a merge-and-split fashion (checks the compatibility between the different nodes), and finally attempts to approximate the continuous dynamics by applying dedicated probability density functions (from the available knowledge one has on the system). Unfortunately, it relies on a lot of information about the system (the components and their different events, the data distribution shapes, etc.); therefore, this technique might achieve great results, it is not merely generic, and it is hardly adaptable to any new plant without a detailed and strong upstream investigation.

Another work that is worthy of attention, on the fault detection domain, is by [45], who aimed to identify unobservable behaviors. Considering that the system events and regular states are known, it is possible to link them within a Petri net, forming the system’s automaton in the Petri formalism. This work proposes following the evolution of the system under consideration and mutating its Petri net to take into account any unforeseen and/or unobservable state (behavior). This method is data-driven and updates the system’s automaton in order to refine it until no more events appear. Unfortunately, it relies on an initial automaton (to be updated and refined), requiring an upstream investigation, which hampers its universality capabilities.

A different approach was proposed by [37]; they introduced the COMMAS method, which uses a hidden Markov model to model a system and then uses it for fault diagnosis and anomaly detection. They worked on a real set of electrical transformers they knew to be fault-free and trained their learning on them. The model is then used as a reference for fault detection on any unknown (same nature) transformer. The use of several healthy transformers aim to obtain a statistical model, since two transformers (or systems, more generally) could have the same references or patterns but different operating areas, due to their intrinsic differences, as small as they might be. Based on another formalism (the hidden Markov model), this method aims to obtain a model one more time, but it requires previous knowledge anew, especially concerning the system’s health.

More recently, and directed more toward machine leaning, reference [38] proposed using data-driven clustering in order to classify the data given by a certain sensor, to gather them into compact groups, assumed to be representative of its different operating areas. Then, they used local probability density distributions to refine the clusters. While this work is not inspired by the above-mentioned reference, the methods are similar, as the philosophies are similar: using machine learning and data-driven clustering to identify the operating areas, especially the self-organizing maps, which are very popular, since they are efficient, accurate, and resource-saving. Whilst the first step of both methods is the same (clustering), their respective goals differ, since we apply clustering to the plant’s feature space, not only to its sensors. Moreover, we propose a methodology to assess the quality of the clusters and on how they represent the plant’s demeanor. Intrinsically, our work and the above-mentioned one are more complementary rather than competitive, as evidenced by their previous paper [39].

In [40], the authors also worked on unsupervised anomaly detection (one could also have a look at [41] for unsupervised anomaly detection within an Industry 4.0 context), but using deep learning, variational modal decomposition, and convolutional neural networks. The main drawback with deep learning is the heaviness of the algorithms, for they require large amounts of data. They are generally very accurate, but also very specific, very long to train, and the collection of so large amount of data could be hindering.

The last work that needs to be discussed is the original paper in which the present article constitutes an extension [42]. We introduced a new methodology to automatically identify and characterize the regular—and irregular—behaviors that a system, comprising a set of sensors, can enter in. To do so, we used machine learning-based data-driven clustering algorithms, namely the self-organizing maps (SOMs) and the kernel k-means (cf. Section 2), so as to gather the data of the system’s feature space in compact groups; assuming the behaviors are quite different from one another, and that the different sensors involved take typical and distinguishable values, it is not a strong assumption to assimilate every cluster to a regular behavior. In order to automatically estimate the relevancy of the clusters, we proposed computing some quantifiers on each, comparing them with each other and to the whole database, so as to automatically decide on the confidence one can have on each: if a cluster has good quantifiers, it is assimilated to a real behavior; otherwise, it is seen as problematic, and further investigation is required. We applied this methodology on real Industry 4.0 data; that allowed us to identify four or five (depending on the clustering technique used) real behaviors within the feature space, corresponding to the real states of the plant (power on and shutdown phases, the steady state, plus one or two less regular behaviors, cf. Section 3).

This present extended version of that previous work mainly consists of specifying the subtleties of our approach, applying them on new data, and finally in adding two new quantifiers to assess the universality of that method, so as to assess if the initial quantifiers we used were representative enough in a real Industry 4.0 context.

Therefore, this article is organized as follows: Section 2 presents the materials of this study, i.e., the clustering techniques, the different quantifiers, and the datasets; in Section 3, all of these tools are applied to real Industry 4.0 data, so as to study and discuss the relevancy of our approach; finally, Section 4 concludes this paper and presents our future works.

## 2. Materials and Methods

### 2.1. Clustering

Space partitioning is a common task in machine learning and data mining [46,47,48,49,50,51]: it offers some clues on the topology of the space and on how the data are related. For decades, several clustering approaches have been developed, mainly based on statistical studies of the data. Some of them are completely *unsupervised* (no previous knowledge required) while some are *supervised* (some knowledge required). The latter generally achieve higher accuracies in the classification process, especially in determining the cluster boundaries; the former are generally used in pre-processing to obtain the data prepared for following steps—either supervised learning or expert-based and case-specific applications.

The main purpose of clustering is to identify compact groups of points [52]. They are generally parametric, i.e., they require some manual meta-parameters for processing. Some algorithms try to estimate their optimal values, but since they highly depend on the data, there exists no perfect way to obtain the optimal parameters suitable to any case.

The main clustering methods can be gathered in three groups: *statistical*, *fuzzy*, and *support vector* [53]. The statistical algorithms draw several points (or the whole database) and adapt a topological figure (similar to a grid) by comparing each one to some other points of reference [54,55]; the fuzzy algorithms are extensions of the previous ones, by letting a point belong to several clusters, allowed by a dedicated formalism [56,57]; support vector is a supervised class of clustering algorithms, which is dedicated more to the estimation of the boundaries of the clusters rather than the identification of the clusters themselves [58]. Since we are only focusing on blind knowledge extraction, we will only introduce the first category, i.e., the statistical and unsupervised clustering approaches.

In the following, let $D={\left\{x_{i}\right\}}_{i\in [\;[1,N]\;]}$ be the database to cluster, with *N* the number of data instances, and let d refer to a distance (Manhattan, Euclidean, etc.).

The most common clustering algorithms are:**K-means** [54]: one of the most common and simplest unsupervised methods. It is a recursive algorithm that aims to identify the clusters by estimating their respective means. The idea is to identify *K* clusters (*K* is a manual parameter) by randomly drawing *K* points, consider them as the centers of the clusters, and then associate any point of the database to the nearest of these *K* points. Once the data are clustered, the *K* initial means are updated as the means of the newly built clusters. This procedure is repeated until some criteria are satisfied. Let $C_{k}^{\left(t\right)}$ and $m_{k}^{\left(t\right)}$ be, respectively, the *k*th cluster itself and its mean at iteration *t*. The clusters are formed following (Equation 1), and their means are updated by (Equation 2).
(1)$$C_{k}^{\left(t\right)}=\left\{x\in D:d\left(x,m_{k}^{\left(t\right)}\right)=\min_{i\in [\;[1,K]\;]}d\left(x,m_{i}^{\left(t\right)}\right)\right\}$$
(2)$$m_{k}^{\left(t+1\right)}=\frac{1}{\left|C_{k}^{\left(t\right)}\right|}\sum_{x_{i}\in C_{k}^{\left(t\right)}}x_{i}$$**Self-organizing maps SOMs** [55]: an unsupervised clustering method that has proven itself through time. It can be seen as a topological *K*-means: the means are linked to each other, somehow forming a grid, whose nodes are the *K* centers of the clusters. When data *x* is drawn, the node’s patterns are hierarchically updated: the nearest cluster’s pattern is the most highly updated (attracted by the data); it is called the best matching unit, and is denoted as $k^{*}$ in the following. This update is then propagated through the whole grid, from a node to another; the farther from the data the node is, the less it is updated. If we ignore the notion of neighborhood, we obtain the *K*-means anew. The neighborhood linkage is given by (Equation 3), and the node’s pattern update by (Equation 6). The learning stops after $T_{max}$ iterations.
(3)$$h_{k}^{\left(t\right)}\left(k^{*}\right)=\exp\left(-\frac{d^{2}\left(k,k^{*}\right)}{2{\sigma^{\left(t\right)}}^{2}}\right)$$
where $\sigma^{\left(t\right)}$ is the neighborhood rate at iteration *t*, which aims to decrease the neighborhood impact through time, defined by (Equation 4), where $\sigma_{0}$ is its initial value (t=0).
(4)$$\sigma^{\left(t\right)}=\sigma_{0}\exp\left(-\frac{t}{T_{max}}\right)$$The learning rate $\epsilon^{\left(t\right)}$ at iteration *t* aims to decrease the learning of the grid through time (to avoid oscillations), and is defined by (Equation 5), where $\epsilon_{0}$ is its initial value (t=0).
(5)$$\epsilon^{\left(t\right)}=\epsilon_{0}\exp\left(-\frac{t}{T_{max}}\right)$$Finally, the *k*th cluster’s pattern, here represented by an attraction coefficient, called weight $w_{k}^{\left(t\right)}$ at iteration *t*, is updated by (Equation 6).
(6)$$w_{k}^{\left(t+1\right)}\left(x\right)=w_{k}^{\left(t\right)}+\epsilon^{\left(t\right)}\times h_{k}^{\left(t\right)}\left(k^{*}\right)\times \left(w_{k}^{\left(t\right)}-x\right)$$

Clustering approaches differ due to their complexities, their application contexts, or their purposes. Each aims to satisfy certain requirements, and there is no perfect or general solution. As we wanted to dig into unknown data to extract any possible knowledge, we relied on unsupervised clustering only, since no specific knowledge was required.

Moreover, following the results obtained in [42], we limited ourselves to the self-organizing maps, for they are very accurate and resource-saving. In that previous work, we obtained better results with the SOMs than with the (kernel) k-means; therefore, they added no true new relevant information. Thus, there is no true reason to focus on them more than for pure comparison, which was already done in that previous work.

**Summary** **1.***Clustering consists of grouping data into compact and homogeneous groups (according to any criteria). It is usually used as an unsupervised pretreatment for further supervised learning: it helps understand the data, or simply automatically label them (classification). Several families exist, among which the statistical, the fuzzy, and the SVM. In this paper, we only focused on the first category, with the self-organizing maps SOMs, an efficient and accurate unsupervised method able to separate even nonlinear datasets*.

### 2.2. Quantifiers

Obtaining the clusters is one thing; asserting their representativity is another. Indeed, most space partitioning and clustering algorithms have recourse to some meta-parameters, especially the number of clusters to build: for instance, it is the *K* of the *K*-means, and the map’s size of the SOMs. As a consequence, one generally needs to have access to some preliminary knowledge about the data so as to correctly set these meta-parameters, which is highly prohibitive in a data mining, blind knowledge extraction context.

However, depending on the clustering method, the impact of these meta-parameters may greatly differ: for instance, the *K*-means will build *K* clusters, all being nonempty by construction; on the opposite, due to the global linkage between the nodes of the SOM, some of its clusters might comprise very few data, or even be totally empty. As a consequence, if the number of clusters set for the *K*-means is higher than the number of real behaviors, some of them will inevitably be broken into several, smaller clusters (i.e., the data of a unique behavior scattered within several clusters); similarly, if *K* is lower than the number of real behaviors, some overlaps (i.e., the data of several behaviors gathered within a unique cluster) would be inevitable. Therefore, the number of clusters *K* is very important for the *K*-means, and will directly drive the accuracy and representativeness of the results. This is a little different for the SOMs: even though the behavior overlaps cannot be avoided when using a too small SOM (grid size lower than the actual number of real behaviors), the behavior splitting effect can be limited when using a SOM wider than required, since some clusters can be fully empty. Even though that does not discard the importance of an accurate setting of the grid size, a wrong choice for it will likely have a lower impact of the results than a wrong choice for *K* of the *K*-means.

In any case, correctly setting the meta-parameters remains important; nevertheless, in a fully data mining approach, very general knowledge should be assumed, and a logical but still-general choice must be made for these meta-parameters, with no evidence that the values set for them are the optimal ones. Consequently, it is important to be able to automatically and blindly assess the cluster quality, so as to know if we can trust the clustering or if some refinement should be required, or even if the full clustering should be discarded in case the clusters are not representative of the real behaviors.

In order to quantify this fact, we propose to automatically characterize the clusters through some relevant and representative metrics. Here comes the problem of the threshold: for a given cluster, which value means a “good” or a “bad” quality? The problem with a threshold is that it is generally a fixed value, set once and for all at the very beginning of the algorithm. As a consequence, we are replacing a knowledge by another, the number of behaviors (grid size) by a manual threshold. To avoid that, we can make a small assumption: at least one of the real behaviors is correctly identified and isolated by clustering; therefore, this cluster should obtain the best properties according to the different metrics (introduced thereafter) and, thus, can consequently be used as a reference—a threshold whose value is dynamically set by the results of the clustering.

This is not a so strong assumption though, especially when working with a large grid, since we can assume that at least one behavior stands out from the others; and if not, this reference can be seen as a kind of random threshold, which may be adapted if necessary in case no cluster seems to be of good quality (all poor metrics). Once clustered, quantified, and the best one identified (according to the following metrics), we compare the other clusters to each other, especially to the so-called reference, which allows to automatically detect and characterize the “problematic” clusters, i.e., those with the poorest properties. Thereafter, they can be processed anew: splitting, merging, or outright discarding.

The quantification metrics can either rely on the ground truth (if available), and are called “extrinsic metrics”, or can be totally blind, based on no knowledge, and are called “intrinsic metrics” [59]. Thus far, we have oriented our methodology toward blind data, assuming nothing about them; in such a context, only the last type of quantifiers can be used, i.e., those relying on no specific information about the data, especially not their ground truths. We start with a statistical comparison of the clusters with the Kolmogorov–Smirnov test, so as to obtain some insight into the data distribution of every cluster, and we then apply three intrinsic quantifiers based on the cluster properties: the repartition of the data with the silhouette coefficient, their scarcities with the average standard deviation, and finally their compactness with the density.

**KS test** [60,61]: an empirical test based on the study of the repartition of real data, the Kolmogorov–Smirnov (KS) test compares the data distributions of two datasets. Assuming a real system S tends to follow a given cumulative distribution function (CDF) $F_{S}$, the dataset $D={\left\{x_{i}\right\}}_{i\in [\;[1,N]\;]}$ recorded during a given window of time should coarsely follow the same distribution, with *N* being the number of samples. Since $F_{S}$ is a CDF, it is also the probability that a data $x_{i}\in D$ is higher than a value *x* (Equation 7).
(7)$$\forall x\in R,\;F_{S}\left(x\right)=P\left(x_{i}\leq x\right)$$As a consequence, it is possible to estimate the empirical probability ${\hat{F}}_{S}$ of appearance of a value *x* by counting the real data whose values are actually higher than *x*, and dividing by the size *N* of the database (to normalize the value), as defined by (Equation 8).
(8)$$\forall x\in R,\;{\hat{F}}_{S}\left(x\right)=\frac{1}{N}\sum_{x_{i}\in D}H\left(x_{i}\leq x\right)\;\mathrm{with}\;H\left(x_{i}\leq x\right)=\left\{\begin{array}{cc} 1 & \mathrm{if}\;x_{i}\leq x\\ 0 & \mathrm{otherwise} \end{array}\right.$$This estimate is motivated by the law of large numbers [62]; indeed, if the duration of the recording and the number of samples are infinite, ${\hat{F}}_{S}$ will converge towards $F_{S}$ (Equation 9).
(9)$$\lim_{N\to \infty }{\hat{F}}_{S}=F_{S}$$The final KS test is defined as the absolute maximal distance from the real CDF $F_{S}$ and any estimate ${\hat{F}}_{S}$, as defined by (Equation 10).
(10)$$\mathrm{KS}=\max_{x}|F_{S}\left(x\right)-{\hat{F}}_{S}\left(x\right)|$$The KS test was originally designed to compare how far from the real distribution $F_{S}$ any other, empirical distribution ${\hat{F}}_{S}$ is. This can be used to compare a prediction to the true distribution of the system, and estimate its accuracy; unfortunately, this requires having access to the analytical distribution $F_{S}$, which is often hard to build, and totally prohibitive in a data mining context. As a consequence, it is barely applicable to our case; nonetheless, it can be used to compare two empirical distributions, so as to statistically estimate how close from one another they are. Ironically, this is not the KS itself that interests us here, but its reverse: the behaviors of a system should be very different, and follow very distinct data distributions; therefore, it is possible to estimate how different and clearly separated the different clusters are from each other. In other words, if the clusters depict clear and unique behaviors, their distributions should be as clear and unique; on the opposite, if a unique behavior has been split into several clusters, they should follow very near distributions, and their KS scores will be very high. The problem with the KS test is that it is normally designed to operate in a 2D-space; since we are working in a multi-dimensional space, we will “simply” compute the KS test along every dimension, and then average all of the dimensions to obtain a scalar score. Another drawback is that it is a binary comparator: the clusters must be compared pairwise, and the local scores must be merged in some fashion.**Silhouettes** [63]: the first attempts to blindly assess cluster qualities were based on a dynamic comparison of the data inside and between the different clusters. From the Dunn index [64] to the silhouette coefficient [63], and passing by the Davies–Bouldin index [65], the main idea was the same: for any given data, on the one hand, compare it to that of the same cluster as its, and on the other hand, contrast it with any other data not belonging to its cluster; only their respective formalism slightly differs. For instance, the most recent (but quite old) silhouette coefficient proceeds in three steps: (1) for any data $x_{i}$, compute the average distance $\mathrm{avg}(x_{i})$ between it and its neighbors within its same cluster $C_{k}$ (Equation 11); (2) for every other cluster $C_{k^{\prime }}$, compute the mean distance between $x_{i}$ and the data belonging to $C_{k^{\prime }}$: the minimal value among them is the dissimilarity $\mathrm{dis}(x_{i})$ of $x_{i}$ (Equation 12); (3) compute the silhouette coefficient $\mathrm{sil}(x_{i})$ of $x_{i}$ by subtracting the average distance $\mathrm{avg}(x_{i})$ from the dissimilarity $\mathrm{dis}(x_{i})$, and divide this value by the maximum among both measures (Equation 13).
(11)$$\begin{array}{cc} \forall x_{i}\in C_{k},\;\mathrm{avg}\left(x_{i}\right) & =\frac{1}{|C_{k}-1|}\sum_{x_{j}\in C_{k}\backslash \left\{x_{i}\right\}}d\left(x_{i},x_{j}\right) \end{array}$$
(12)$$\begin{array}{cc} \forall x_{i}\in C_{k},\;\mathrm{dis}\left(x_{i}\right) & =\min_{\begin{array}{c} k^{\prime }\in \left[\;\left[1,K\right]\;\right]\\ k^{\prime }\neq k \end{array}}\left\{\frac{1}{|C_{k^{\prime }}|}\sum_{x_{j}\in C_{k^{\prime }}}d\left(x_{i},x_{j}\right)\right\} \end{array}$$
(13)$$\begin{array}{cc} \forall x_{i}\in C_{k},\;\mathrm{sil}\left(x_{i}\right) & =\frac{\mathrm{dis}\left(x_{i}\right)-\mathrm{avg}\left(x_{i}\right)}{\max\{\mathrm{dis}\left(x_{i}\right),\mathrm{avg}\left(x_{i}\right)\}} \end{array}$$The silhouette coefficient takes into account the relation between the data, both inside and outside a cluster. The avg measure captures the compactness of the clusters, i.e., how close the data within are: the lower, the more compact, with value 0 as its best case (data overlay); the dis measure depicts how distant the clusters are from one another: the higher, the more distant and, thus, the better (clear borders). The final sil coefficient represents how well each point has been classified, which can been seen as a regulated dissimilarity measure, whose score is reduced by the average distance as a penalty: the coefficient is higher in case the data of a given cluster are very close (avg) and that cluster in question is well-separated from the others (dis), and vice-versa. Dividing by the maximum among the two local scores only aims to normalize the final score, so as to ease the interpretation of the silhouette coefficient. It takes a value between −1 and +1, with −1 meaning that data $x_{i}$ is on average nearer to the point of a different cluster than that of the cluster it is actually in, and that it has probably been wrongly classified; and value +1 meaning that $x_{i}$ overlays its neighbors of its cluster (maximal compactness), and is far from the other clusters; as a consequence, the nearer to +1, the better. This measure is very representative of the classification, but its main drawback is it is very heavy to compute: it is not suited for large databases. Moreover, it does not consider the cluster size: a cluster comprising a single value, but distant from the others (i.e., an outlier), will obtain the optimal value +1: with no more attention put on it, it might be assimilated to a real behavior, whereas it is in fact a “simple” outlier.**AvStd** [66] In statistics, one of the most common tools is the standard deviation, for it informs about data scarcity [67]. It is computed along a given axis; with N axes, there are N values, one per axis. It is generally harder to handle a feature vector than a single scalar, especially when considering other scalar metrics. As a consequence, reference [66] naturally proposed to compute this metric along every dimension, and then fuse the vector’s components in a representative fashion. It investigated several ways for the merging: minimum, maximum, mean, etc. It ended with the conclusion that the most representative and the most universal is the average of all these standard deviations; hence, the name of this metric: average standard deviation, or AvStd. Based on a statistical study of the data, it has the advantage to diminish the impact of the outliers. The drawback is that it is essentially an indicator of data scattering. The standard deviation is computed feature-by-feature, and the AvStd measurement is the average. Let $m_{k}$ be the mean of the *k*th cluster, defined by (Equation 2), whose data are defined by (Equation 1). The standard deviation $\sigma_{k}$ is defined by (Equation 14).
(14)$$\sigma_{k}^{\left(i\right)}=\frac{1}{\left|C_{k}\right|}\sum_{x\in C_{k}}{\left(m_{k}^{\left(i\right)}-x^{\left(i\right)}\right)}^{2}$$
where i∈[[1,n]] is the current feature, with *n* the feature space’s dimension. The AvStd measure of cluster *k* is finally given by (Equation 15).
(15)$${\mathrm{AvStd}}_{k}={\bar{\sigma }}_{k}=\frac{1}{n}\sum_{i=1}^{n}\sigma_{k}^{\left(i\right)}$$**Density** [31] In physics [68,69,70], the density of an entity is defined as the ratio of the number of items contained within to the volume it occupies; more specifically, to be a true density, this value must be divided by a reference, since a density has no unit. The question is to know what a volume is; indeed, in dimension 3, there is no ambiguity, but this notion becomes more blurred when the dimension increases. To answer that, we propose using the hyper-volume theory, which provides N-dimension equivalents to the regular 3D volumes: for instance, a 3D sphere becomes a ND hypersphere. Therefore, we propose representing a cluster by a hypersphere and, thus, assimilate both volumes: the cluster’s volume is that of the smallest hypersphere containing all of its data points. It remains to evaluate the cluster’s span; the most natural estimate would be the maximal distance separating two of its points, but it is quite heavy to compute (complexity of $O(N_{k}^{2})$, with $N_{k}$ being the number of points within cluster $C_{k}$). To compensate that, we decided to find the maximal distance between any point and a fixed reference (for instance the cluster’s mean or its pattern), which greatly reduces the complexity ($O(N_{k})$), and to double it so as to estimate a hypersphere containing all of the points. The first solution, the maximal distance between two points, leads to the smallest hypersphere containing the cluster, whilst the second approach, the double of the maximal distance from a reference, gives a higher estimate of the cluster’s volume, but is easier to compute, and is centered around the cluster’s pattern. Moreover, a cluster’s pattern is mostly a region of influence, which only depends on the observed data (data belonging to a given class could be outside the identified cluster’s borders trained upon the available observations): a higher estimate is not so problematic in that case. The cluster’s span is therefore given by (Equation 16).
(16)$$s_{k}=2\times \max_{x_{i}\in C_{k}}\left\{d(x_{i},m_{k})\right\}$$
and, according to [71], the hypersphere volume is given by (Equation 17).
(17)$$v_{s}^{\left(n\right)}\left(r\right)=r^{n}v_{s}^{\left(n\right)}\left(r=1\right)=\frac{r^{n}.\pi^{n/2}}{\Gamma \left(\frac{n}{2}+1\right)}$$
where *r* is the hypersphere radius ($s_{k}$ in our case), and Γ is the Gamma function, defined by (Equation 18).
(18)∀z∈C:Re(z)>0,Γ(z)=∫0+∞xz−1e−zdxThe density $\rho_{k}^{\left(n\right)}$ of cluster *k* is finally computed by dividing the number $N_{k}$ of data instances contained within by its volume $v_{s}^{\left(n\right)}$; the density is given by (Equation 19).
(19)$$\rho_{k}^{\left(n\right)}=\frac{N_{k}}{v_{s}^{\left(n\right)}\left(s_{k}\right)}$$In [31], we tested this density-based metric on both academic and industrial datasets; it proved to be very reliable to characterize the clusters and to evaluate their qualities. It is quite efficient and representative of both the outliers and the number of data contained within. In our previous study, we endeavored to compare the densities of the different clusters, but in this present article, we will “normalize” them by the value of the database itself, since using a reference sanctifies the notion of density.

It is now time to introduce the databases upon which we will apply our methodology, in order to assess it, equipped with self-organizing maps to break (cluster) the feature space into compact and homogeneous pieces, and with four quantification metrics to automatically characterize the so-built clusters.

**Summary** **2.***Clustering is a powerful (albeit uncertain) tool, for it operates in a fully blind environment. Assuming we do not have access to the ground truth or the real classes, we need to assess whether the obtained clusters are trustful, or simply representative of anything; to do so, we propose to consider four different quantifiers, summarized within Table 1, which inform about how they are representative of the subregions of the feature space, subregions that may be assimilated to real behaviors of the system*.

### 2.3. Industry 4.0 Datasets

This papers takes place within the HyperCOG project, a European Research project (cf. section Funding) focusing on Industry 4.0 and the cognitive plant of the future. With 14 partners around this unique goal, some researchers work on issues pertaining to the hyperconnectivity of such a plant [72], whilst others (including us) work on knowledge extraction and conceptualization [31,42]. In this context, one of the companies we are in partnership with is Solvay^®^, specifically their factory located at La Rochelle, France, a chemistry plant specialized in the manufacturing of rare earth (RE) specialty products.

A detailed description of their processes is not mandatory, nor even necessary, to study an automatic knowledge extraction procedure using unsupervised machine learning-based clustering methods. The reader should only know that the RE extraction procedure comprises several steps, leading to a large number of parameters and, therefore, of sensors. Nonetheless, according to the Solvay experts, some are more relevant than the others; indeed, during the RE extraction, some steps are essential, and a small local disturbance might have a very significant impact and totally disturb the whole procedure. In the following, for representativeness concerns, only these most important sensors are depicted. Consequently, the feature space is the *n*-dimension Euclidean space generated by the basis whose orthogonal axes are the features of the most important *n* sensors.

We should also specify that the RE extraction is binary; let us consider for instance some raw material composed of four different REs: the idea is to separate the four initial ones into two subsets of two REs, and repeat this separation procedure for each subset. As a consequence, the extraction is composed of three stages, one for the two subsets of two REs, and two others for each subset. Such a stage is called a “battery”.

Solvay experts provided us with the data recorded in different batteries of the plant; whilst the original work [42] dealt with only one of them, we will consider another one in order to assess our results on more data and to emphasize the universality of our methodology (no case dependent). We will refer to the first process as “battery 1” or “B1”, and the second process as “battery 2” or “B2”. Note that these two processes are actually closely related, but not directly connected.

Let us define $D_{1}$ and $D_{2}$ the databases of B1 and B2, respectively; their data were recorded at the same times, every minute, over a month and a half, for a total of seven running weeks. Typically, Solvay’s plant runs for several consecutive days, and is shut down for a while before starting a new working week; both databases comprise seven sets of restart, steady state, and shutdown procedures. Due to a short-time failure in battery 2 (sensors temporarily offline), some instances are not correctly exploitable, and have been removed; therefore, $D_{2}$ is slightly smaller than $D_{1}$, with, respectively, 63,715 and 65,505 data points. Finally, both databases comprise a hundred sensors, but many are strongly linked to each other and are not all equally representative; $D_{1}$ contains eight very essential sensors, whilst $D_{2}$ relies on twelve major sensors. The number of sensors serves as the dimension of the feature space. The evolution over time of the eight sensors of $D_{1}$ is depicted in Figure 1, and Figure 2 represents the twelve sensors of $D_{2}$. Note that the data are normalized in order to avoid that a dimension weighs more than the others due to its only scale.

On almost every sensor of both figures, two motifs are easily distinguishable, for they form compact groups, which repeat over time. They are circled in green and red on both figures for clarity. As a general rule, with a living and evolving system, anyone would expect at least two states: running and shutdown. On both figures, the two circles are coarsely those states: the red ones represent the sensor states when the battery is shut down, whilst the green ones are those when the battery is running.

An overall view of all the sensors of a battery, displayed side-by-side, makes it possible for a human being to decide on the number of behaviors—or at least an idea of it. This is not as easy for a computer; indeed, considering a set of *N* sensors, a typical scenario may appear: *N*-*M* sensors take “regular” values, while the *M* last sensors do not, with *M* being a strictly positive integer less than *N*. Such a scenario typically consists of a local failure of some sensors, or a beginning of an upset. One example is visible on Figure 2, where a group of data of sensor 7, circled in pink, takes an “irregular” value, never seen before, whereas such an event is not visible on most of the other sensors; this was due to a real local failure in battery 2, which affected some sensors first—especially sensor 7, and which was prevented from causing greater casualties within the battery by the expert operators.

It is from there that (automatic) behavior identification and characterization becomes interesting: pointing out the regular behaviors of a system and isolating the irregularities, either by extracting differing and salient groups of data, or by comparing all groups to those deemed as “regular”—the irregularities are those left with no correspondence.

Now the databases are introduced, we will endeavor to extract the different behaviors of each battery by following the methodology we presented in Section 2.1 and Section 2.2. Finally, let us say a few words about the quantification of both databases. Even though neither the silhouette coefficient nor the KS test can be computed for a sole group of data, as there is no comparison possible (and a comparison between a database and the other one is not representative with these metrics), the density and the AvStd measures can; they are gathered within Table 2.

These values are given for comparison purposes: they will be used as references for the clusters given by the SOM in Section 3. Even if the intrinsic value of the density has no meaning by itself, it can nonetheless state that the first database $D_{1}$ is slightly more dense than the second $D_{2}$, mostly due to a greater number of points. Assuming that is true, one could argue that their respective scarcities are similar, if not statistically identical. That being said, the standard deviations, along every axis (sensor), indicate that $D_{2}$ is slightly more compact than $D_{1}$, with very slightly different minimum and average standard deviations along the axes (note their average is the true AvStd measure).

**Summary** **3.***In this paper, we will deal with two databases, both issued from a real industrial plant, currently mutating toward Industry 4.0. These two databases correspond to two different parts of the system, called “battery”; these two batteries are related but not directly linked. They both comprise about 65,000 data samples, recorded over seven work weeks, and a hundred sensors; nonetheless, only some of them will be considered for simplicity: eight sensors of battery 1 and twelve for battery 2. In both cases, we apply a SOM in order to attempt to isolate the real behaviors of the system, assimilated to compact, dense, and homogeneous feature space subregions. In both cases, three noticeable behaviors stand out: the steady state, the shutdown, and some intermediate states, among which, an anomaly. We will endeavor to identify and characterize these three regions*.

## 3. Discussion of the Results

Now that everything is in place, it is time to apply the SOMs on both databases to study the feasibility of how to automatically extract the different behaviors of each battery.

Let us begin with setting the grid’s size; since we are working in a data mining context, we should assume nothing but general information about the batteries. As a general rule, as mentioned above, any real system should behave in at least two different manners: running and shutdown. Assuming these two behaviors can slightly overlap (especially right before the beginning of a phase or right after its end), a map of four or five nodes sounds reasonable for the regular behaviors; two or three additional nodes should also be considered for the irregularities (anomalies) plus one or two to absorb the possible outliers. As a consequence, with only very general considerations, a map of about six or seven nodes should be chosen with regard to any real system; less risks leading to an overlap between different behaviors, whereas more might lead to empty or inconsistent groups of data (the map would be too stretched). With respect to all of these considerations, the final choice is thus made on a map of nine nodes (topology 3×3) for both batteries.

Let us finally define the color scheme used all over this section, for both figures and tables. For consistency, the clusters will follow a unique color correspondence defined as of Table 3; this means that when a cluster will be referred to as “cltX” in a Table, the corresponding color will be used to display its data in the corresponding figure. Note that the colors are only here for clarity, and mean nothing by themselves (they could have been exchanged without modifying anything in the following).

### 3.1. Battery 1

Let us begin with battery 1 and its eight sensors, comprising 65,505 timestamps each. As mentioned in Section 2.3, it contains two main groups of data, plus a few possible upsets: a 3×3 SOM should be enough to include and represent all the real behaviors (even though it could be too many, but it is the opportunity to assess our methodology in bad conditions).

#### 3.1.1. Preliminary Qualitative Study

Let us start by applying a self-organizing map to battery 1’s database. It is important to remember that it is a statistical approach: therefore, the clusters will slightly mutate from a round to another; indeed, since the SOMs greatly depend on their initial conditions (the very first points to be drawn serve as the very first node’s patterns), the final clusters may change a little. That being said, the main regions should not be impacted that much; experimentally, many attempts were performed, leading to three possible configurations: two, three, or four main groups of data appear, as depicted in Figure 3. After a few dozens tries, we did not obtain a different combination.

With battery 1, three configurations stand out. The first (leftmost column on Figure 3) is composed of two very large clusters, which means that two true behaviors have been identified—it the best case for this battery. The second (column in the middle) saw the compact blue group of data (shutdown state) be broken into two halves, corresponding to the real steady state of the battery in blue, and the start-up procedure in red. The last configuration (rightmost column) saw both groups be divided into two halves, mostly due to the high tolerance—and scarcity—of sensor 8: with four main clusters, for only two real behaviors (in this case), this configuration is the worst.

These three outputs are highly linked to one another: the case “2 clusters” is the best because it represents the two main states of the plant, i.e., running and shutdown. The two other cases are simply one or both of these clusters which was/were broken into two halves, due to some scarcity in the sensor data. What is comforting nonetheless is that the overall clusters remain globally the same; for instance, the blue cluster in “2 clusters” is cut in two halves (blue and red data) in “3 clusters” and “4 clusters”. This is exactly the same for the green cluster, which can be split into two sub-clusters (green and orange) across the attempts. Moreover, a blue cluster (or a pair of a blue and a red clusters) on Figure 3 substantially corresponds to the motif circled in red on Figure 1; similarly, a green cluster (or a pair of a green and a red clusters) corresponds to a motif circled in green in the respective figures. Note that even though the clusters were colored so as to match the color scheme of the circles of Figure 1, they were obtained automatically by a SOM.

As a general rule, the clusters remain the same, only cut in half. This low scarcity in the results is very reassuring, for it shows that they are consistent, and that the study we will carry on only one try would likely be similar for any other. As a consequence, we will conduct our quantitative study in one case only, assuming similar results would be obtained when considering another combination. In order to remain as general as possible, we will consider the worst case; indeed, who can do more can do less, if our methodology proves to be trustworthy in bad conditions, it would also be trustworthy in good conditions, even with better results! For this reason, we will only consider the case “4 clusters”.

The eight sensors of that configuration are depicted in Figure 4; in that figure, four main groups of colors stand out: blue, red, green, and orange. The first two correspond to the shutdown state, more precisely the real shutdown in blue, and the start-up in red; the last two correspond to the real steady state, cut in two halves due to a tolerance in sensor 8—actually, both should be merged. A fifth color appears, the cyan, at the very bottom of the blue cluster, corresponding to the initiation of the shutdown procedure.

Even though there are nine colored clusters in the figure, six are very distinguishable: clt1, clt2, clt3, clt4, clt6, and clt9; the three remaining clusters are almost empty—reason why they cannot be seen with ease—and could be assimilated to true outliers. The fact that some nodes are almost empty emphasizes the conviction that the grid’s size (nine nodes) was a little too large; as a consequence, such a grid seems perfectly suitable for this database. Concerning the six nonempty clusters, four are very large (several thousands of points), and the two last ones are small, more representative of local behaviors.

Actually clt1 is the shutdown state of the battery, whilst clt4 and clt9 are two other parts of it; a unique state is split into three parts. More precisely, when one digs into the timestamps and links them with the plant’s running agenda, the blue values appear when the plant is totally shut down, i.e., several hours after the shutdown procedure has been initiated, and several hours before any new action has been performed: it is the “real” shutdown state of the plant. The red data appear a few moments after a wake-up command has been ordered, at the very beginning of a new work week: it is the “real” start-up phase. Finally, the cyan data always appear at the end of a work week: it is the shutdown order sensu stricto. As a consequence, even though these three clusters are parts of a unique state, they each might find a semantic identity.

Concerning the three other clusters, clt2 and clt3 are actually parts of the same one, whose data appear at any time within the plant’s working period. There is no real distinction between both except a slight tolerance in sensor 8 leading to the detection of two clusters whilst there is actually only one; hopefully quite rare, as evidenced by Figure 3, this is a limitation of clustering, since it leads to the identification of two behaviors, hardly linkable to one another, for only one real. The last cluster, clt6, is actually a real failure in battery 1, an unforeseen event that forced a manual and premature stoppage of it.

**Summary** **4.**
*Database $D_{1}$ comprises three main regions: steady state, shutdown, and intermediate/ anomaly; these regions are themselves composed of local subregions, such as the shutdown with its three phases: power-off, true shutdown, and start-up. The SOM proved able to detect and separate up to six regions: the three phases of the shutdown, the steady state (unfortunately broken into two parts instead of a unique compact cluster), and an unforeseen and abnormal behavior. The main areas of the feature space were aptly identified and isolated, with few behavioral overlaps; mostly behavioral splits appeared, which were not dramatic since that only meant finer clustering.*


#### 3.1.2. Kolmogorov–Smirnov Test

Now that the qualitative study has been performed, it is time to apply our data mining, blind tools for automatic characterization of the clusters. In this subsection, we will only focus on the KS test, for it is in between a qualitative and a quantitative analyses; the next subsection will only focus on the scalar quantifiers introduced in Section 2.2.

As mentioned earlier, the KS test is a binary comparison between two 2D datasets; since we are working in a space of a higher dimension (eight for battery 1) and the number of datasets (clusters) is higher than only two, we must adapt this test to our needs. For the first point, we need to transform a vector measure into a scalar one; a simple possibility is to deal with the mean of the empirical probabilities, computed along every dimension (sensor); for the second point, when the number of datasets increases, it becomes necessary to compare any possible pairs. By combining both, we end up with a matrix of scalar values, which comprises all the KS tests for any possible pair of clusters.

Following the same color scheme than that defined within Table 3, the empirical data distributions of both the clusters and the database are depicted as in Figure 5, and the mean probability (i.e., the mean of all the sensors of the previous figure) is depicted as in Figure 6.

By analyzing these figures, one may draw three important observations: (1) no cluster follows the same distribution than that of the database; (2) some clusters are more homogeneous than others; (3) there is no pair whose clusters share similar distributions.

The first point means that the database’s distribution is unique, and is probably composed of a mixture of local cluster data distributions, for it globally follows the average of the trends. Moreover, assuming the SOMs globally well separated the dissimilar data from each other, the fact that the database’s distribution is not mimicked by any cluster’s one seems to indicate that some local behaviors actually exist, since there is neither homogeneity nor similarity between the clusters and the database.

The second point is justified by the fact that some distributions have very sudden and extreme variations, indicating that the data are very similar. Let us for instance consider the cyan curve (clt9): for most sensors, it very suddenly variates from null appearance (0.0) to almost all data comprised (1.0). As an example, for the cyan data of sensor 5, there are very few data whose values are higher than 0.05, since the probability of appearance of this value is 0.987; as a consequence, almost all of the cyan cluster’s data have values lower than 0.05, meaning it is very homogeneous (for sensor 5, at least). This is not true for all of the sensors, some of them having much smoother variations; for instance, the red curve (clt4) on sensor 3 has its data values more scattered within this dimension: it is therefore less homogeneous compared to some other clusters (for sensor 3, at least). In reality, the higher homogeneity has a simple origin: it is due to a much lower number of data, since the general observation is that the most homogeneous clusters (pink clt6, gray clt7, olive clt8, and cyan clt9) are also the smallest. This is a drawback of the KS test, since it does not take into account the size of the dataset, contrary to the density and the silhouette coefficient.

The third and last point is the most interesting: no two clusters have the same distributions. This means that all of the clusters are distinct and unique, for there is no redundancy and, therefore, every cluster represents a unique part of the feature space (no bad overlap). In reality, it is a little more complex: some clusters follow very similar distributions within several dimensions, but not within all of them; this is more visible in Figure 6, where some groups of curves appear: blue–red, green–orange, cyan–pink, and perhaps gray–purple–olive. These groups are actually the real behaviors of the system, with, respectively, the shutdown, the steady state, the power-off/event state, and some outliers. The empirical probability shows here that some clusters are actually similar, which seems to indicate that a real behavior has been split into several clusters (which is actually true here). Nonetheless, the distributions are not identical, the reason of this split, since for one or two dimensions (sensors), there are some differences, such as sensor 8 for the orange and the green curves, which are two parts of the same behavior (the steady state), but which actually differ uniquely within this dimension.

Let us eventually compute the KS tests for each pair of clusters, as defined by (Equation 10). To compute a KS test in an 8D space, the empirical probability ${\hat{F}}_{S}$ used here is the mean of that over every dimension. The matrix of all the KS tests is given as of Table 4. Note that the values are represented in percentages (probabilities multiplied by 100) to ease the reading. Note also that the matrix is symmetrical, since the KS test is also symmetrical by itself.

The KS tests globally confirm the observations we draw using the empirical probability distribution graphs: indeed, a high score means that two distributions are highly dissimilar, which is a good thing in our case, since we want to ensure that the SOMs clustered the data into homogeneous and highly distinct groups; however, a low score means that two distributions are very similar, which would indicate a possible issue here, such as a unique behavior split into pieces or an overlap of behaviors by a unique cluster.

From the table, it is possible to draw some observations about the closeness of the clusters: for instance, clt1 is globally far from most of the clusters, except clt4, clt9, and clt6. This is logical, since clt1 is the shutdown state, clt4 is the start-up procedure, clt9 is the power-off order, and clt6 is a bad event, which led to the power-off of the battery: they are all highly similar within the feature space. A similar closeness can be observed between clt2 and clt3, with a low KS score, meaning that these two clusters are quite near in the feature space (they are nearer from each other than from any other cluster), which is actually true, since they are both part of the steady state, cut in halves due to the scarcity of sensor 8.

While adding nothing really new from the manual observations we made previously based on the CDFs, the KS tests automatized the procedure: a low value indicates a possible issue, such as a possible behavior split or overlap, whilst a high value tends to indicate the clusters are very dissimilar, which increases our confidence in the clustering.

**Summary** **5.**
*In summary, concerning the KS test, we were able to manually assess the clustering results (four main regions in the feature space: steady state, shutdown, intermediate, and anomaly) by computing their respective empirical probability distributions; we also did it automatically by using the KS test sensu stricto, where the values directly indicated a possible issue. Nonetheless, we also saw that the KS test could be misled by the cluster size, and it did not help interpret the clusters, it only gave some clues about how well (or badly) separated they were from each other.*


#### 3.1.3. Quantitative Assessment

Let us attempt to assess all of these qualitative, knowledge-based conclusions through the quantifiers we introduced earlier; they have been computed and gathered as of Table 5.

One may wonder if it is possible to decide on the quality—and relevance—of a cluster through the automatic study of some quantifiers; in our case, and without paying attention to our qualitative observations drawn from Section 3.1.1 and from our expert-based knowledge, can we understand that there really are 5–6 distinct behaviors? Let us deal with the quantitative results; we will consider each quantifier separately, draw as much information as possible, and eventually cross-reference their results.

Let us begin with the average standard deviation (AvStd): the smaller the better. Database $D_{1}$ had an AvStd of 0.280, several times greater than any of the nine clusters, which are thus all more compact; this is not surprising though, since this is how the SOMs work: they gather the nearest points from a node’s pattern so as to form compact and homogeneous groups. The maximum value has been achieved by clt8, and the minimum, by clt1, neck and neck with clt6: there is a (normalized) ratio of about 2.5 between the minimum and maximum. With respect to column “$÷\;{\bar{\sigma }}_{\max}$”, and assuming that the most compact cluster clt1 is representative of a real behavior—a necessary assumption for our blind methodology, so as to create an artificial reference, we can argue that the clusters with the closest AvStds to them are also acceptable and representative of real behaviors; as such, clt1, clt2, clt3, clt6, and perhaps clt7 are compact enough to be eligible to the status of real behaviors. Nonetheless, the last one being almost empty, it can be removed: four behaviors are thus identified. In reality, concerning these results, the three main regions of the feature space have been identified: clt1 for the shutdown state, clt2 and clt3 for the steady state, and clt6 for the anomaly. Unfortunately, AvStd missed two clusters: clt4 and clt9, with high values, too high to be considered of good quality; as a general rule, when such a case appears, i.e., with bad quality clusters, they might be worthy of new treatments, either manual or automatic, but required nonetheless. These two clusters were missed for they are a little too stretched over the feature space: they do not form compact groups of data, the reason why the AvStd is a little too high.

Let us continue with the density: the higher the better. That of database $D_{1}$ is 44,401, whilst the maximum among the nine clusters is 29,186, achieved by clt2, for a ratio of 1.52, and the minimum is 5.286, achieved by clt5, for a ratio of 8400. Assuming the database is quite dense, it means that clt2 is also very dense, whereas clt5 is positively not; this is due to the very low number of data in the last cluster. Considering the scale between the maximum and the minimum, i.e., 29,186 to 5.286, for a ratio of 5521, any cluster whose density is, for instance, at most 10 times smaller than the maximum, can be assimilated as real behavior. As such, clt1, clt2, clt3, and clt4 can be assimilated to real behaviors, which is in fact true: the steady state and the shutdown. As a consequence, the density missed clt6 and clt9, but correctly rejected the three others. However, when considering column ”$÷\;\rho_{\max}$”, three orders of magnitude appear: $10^{-1},10^{-2}$, and $10^{-4}$, corresponding, respectively, to the four largest clusters, the two missing, and the three outliers. As such, by increasing the tolerance placed on the difference between the maximal gap in values, the two last clusters may be considered, for they are not the least dense. This quite low density is due to the low cardinality and the scarcity in the data; for instance, with sensor 2 of Figure 4, clt9 comprises two parts: one at the very bottom of clt1, and another that overlaps with the leftmost points of clt2: in reality, this cluster should be split into two halves and redistributed into the two others, which is the reason for this low density, which is not completely wrong here.

Let us finally consider the silhouette coefficients: the closest to 1, the better, the closest to −1 the worse, 0 being a gray area where the data are, on average, at the same distance from their neighbors, and from the points of another cluster. Since there is a silhouette for every point, the 65,505 coefficients cannot be displayed; instead, the minimum, mean, and maximum are represented. Statistically, the most central points of a cluster will obtain better results than the outermost, farther from their neighbors, and nearer to the other clusters; as such, achieving a high max or a very little min has no real meaning by itself, but for emphasis only. Among the column “Mean”, a negative value is often prohibitive, a sign of a very ill-built group of data; as such, clt5 and clt8 can be excluded as of now. On the opposite, clt1, clt2, clt3, and clt6 achieve great mean SCs and, thus, have passed that test of quality, giving similar results, thus far, compared to AvStd. Concerning the three remaining clusters, with quite low SCs, but too high to be rejected, it is hard to decide directly; clt7 has acceptable values, both in max and min, but complementary considering its cardinality, it can be removed. Finally, what about clt4 and clt9? The former achieves a honorable mean of 0.291, with a quite high max of 0.439, and a not so low min of −0.289: it might be considered as representative of a real behavior if the tolerance is decreased. However, concerning the last one, its results are very poor: it cannot be considered as good quality; this is due to the scarcity in this cluster, similarly to the density: it should definitely be split and redistributed. As a consequence, the SCs have allowed to point out four or five clusters (1, 2, 3, 6, 4), depending on our confidence, and to reject three with the intrinsic values only (5, 8, 9), plus another one by considering its cardinality (7). Globally, the three regions of the feature space have been correctly identified.

**Summary** **6.**
*All three quantifiers characterized clt1, clt2, and clt3 as good quality, and as such, they almost certainly represent real behaviors, or at least real regions of the feature space, which might be worth being overlapped or merged. Moreover, clt6 had good results with two of the three methods and an intermediate quality with the third; similarly, clt4 was characterized as dense, but had mixed SCs and AvStd. Due to a weak clustering, clt9 had very poor results with the three methods: it is clear that there is something wrong with it, and is worth being refined. Finally, the three methods aptly rejected the three last clusters. In conclusion, what can be understood is that only one method is not trustworthy enough, and it is better to rely on different approaches, which must then be cross-referenced to validate the results, or to compensate their respective drawbacks.*


#### 3.1.4. Local Conclusion about Battery 1

What are the final conclusions of this study? In short, none of the three quantifiers is suitable to the six main clusters, for there are always one or two with poor results. More than due to a wrong choice of quantifier, this is due to an imperfect clustering. According to the qualitative study of Section 3.1.1 and to the KS test of Section 3.1.2, the clusters were relevant, but this was true at a coarse grain; indeed, if one digs into the clusters, some overlaps or wrong classifications appear, such as the cyan cluster clt9, cut in two parts, one near the blue cluster clt1, and the other one near the red cluster clt4 (visible in most sensors of Figure 4). This can also be drawn from the KS tests of Table 4, where the scores between clt9 and clt1 on the one hand, and between clt9 and clt4 on the other hand, are very low, which therefore indicates a proximity of clt9 with the two others; however, it also indicates that clt9 and clt6 are close, which is not false by itself, but clt6 is worthy of a unique cluster, for it is an event whose values slightly differ from the surrounding ones.

The main limitation of the KS test is that it performs quite badly in a space whose dimension is higher than two, and that it lacks an interpretation capability, since we are working with mean and maximal scores; nonetheless, if one may separate the wheat from the chaff, the KS tests can help point out the issuing clusters, or at least inform about the possible behavioral splits and overlaps. Ironically, this fact clearly stood out via two quantifiers (density and silhouettes); this paves the path for more complex and refined future works. As a result, by cross-referencing the results automatically drawn from the quantifiers and the KS scores, it is possible to increase the confidence one may have in the clustering, and on where to act to improve the results thereafter.

Finally, a more anecdotal result is gathered as of Table 6: the execution times of the four methods, where the notation follows that defined for Table 5, plus the KS tests, where “CDFs” are the empirical cumulative distribution functions, and “Scores” are the KS tests sensu stricto. What is interesting are the orders of magnitude: AvStd is the fastest method, then comes the density, which is two times slower, then the KS test, about thirteen times slower, and finally the silhouettes, with their complexities in $O(N^{2})$, are around 52,000 times slower, for a database of “only” 65,505 data records in dimension 8. In a real-time embedded context, that slowness can be prohibitive; in conclusion, in order to use our methodology, we strongly recommend relying on the first two quantifiers, namely AvStd and density, for they are fast and work very well side-by-side, plus the KS test for cross-validation. The silhouette coefficient might serve for refinement in case of doubt.

### 3.2. Battery 2

In order to assess our methodology, we also tested it on battery 2 to serve as another example to show its universality for automatic behavior identification. We followed the same steps as before, but the other way around this time: we dealt with the quantifiers first and then with the qualitative results so as to avoid any possible bias one could argue. Indeed, our method is based on the automatic analysis of a set of quantifiers; therefore, this test was performed in real conditions, for we assumed nothing before analyzing the metrics, and only after we confirmed the conclusions drawn by a qualitative study.

#### 3.2.1. Quantitative Study

Let us first have a look at the clustering so as to introduce the clusters. Similarly to battery 1, Figure 7 shows the three possible outcomes of the SOMs; notice that the previous color scheme was kept on purpose for consistency, i.e., green and orange for the steady state; blue and red (plus cyan if so) for the shutdown; and pink for an abnormal event. Once more, the colors have no meaning by themselves, they are just set for clarity.

In the figure, only two sensors are depicted for simplicity: sensor 9 at the top and sensor 12 at the bottom. Similarly to battery 1, with database $D_{2}$, three combinations stand out with a nine-node map, with either four or five main clusters. Globally, the same groups of data than that of $D_{1}$ appear: the steady state in green–orange, the shutdown state in blue–red–cyan, and the events in pink–olive. Once more, the two regular events might be cut in two halves, either the steady state (middle and right columns) or the shutdown (left and right columns). In the three cases, a real event was identified in pink, sometimes cut itself in two halves, colored pink–olive (right and middle columns).

As usual, battery 2 behaves in two different ways: steady state and shutdown. These two great regular behaviors are always identified, but often cut in half; sometimes only one of them is split, and sometimes both are. The last case is again the worst, for the consistency and the unity of the clusters is highly decreased; therefore, this is the case we will consider.

This time, let us begin with the quantifiers, to see if we can draw relevant information from blind, cold values gathered in an also blind, equally cold table. Let us consider Table 7, which gathers the results of the quantifiers applied to the case “5 clusters” of Figure 7.

Let us again start with the average standard deviation: that of database $D_{2}$ is 0.267, the lowest 0.044 is achieved by clt1, and the highest 0.152 is achieved by clt6, for a ratio of 3.5. This gap is little greater than for battery 1, which seems to indicate that the delimitation of the clusters might be slightly better. Moreover, there are more acceptable values: clt1, clt2, clt3, clt4, clt6, and perhaps clt5 with a higher tolerance. As a consequence, the acceptable AvStds (compared to the highest or equally to that of the database) seem to indicate that there are between four and five compact clusters and, thus, behaviors, in this battery; notice that the best values are generally achieved by the largest clusters, which means their data are statistically and closely centered around the node’s patterns.

Let us confirm (or invalidate) that with the density: that of database $D_{2}$ is 39,935, the highest 18,216 is achieved by clt1, and the lowest 8.387 by clt8. Compared to either the database or the maximum, five results stand out, for they are at most ten times smaller: clt1, clt2, clt3, clt4, clt6. However, the four others are a thousand times smaller, which is prohibitive, especially clt5: whilst the AvStd was in trouble in regard to deciding whether or not this cluster was of good quality, the density clearly answers no to that question.

What about the silhouettes? In the middle column, i.e., the mean of the SCs, there is no negative and prohibitive value, but some are very small. Actually, except for two or three clusters, namely clt7, clt8, and perhaps clt9, the six others have very good results—at least in an honorable average—especially considering the “Mean” and “Max” columns. According to that metric, there might be up to six or seven distinct behaviors; this is too many though, and the fact that the mean silhouettes are very close for most of the valid results is a bit alarming: there is nothing to draw with certitude. That being said, these observations do not go against that made with the first two quantifiers, emphasizing that clt1, clt2, clt3, clt4, and clt6 are real behaviors on their own; only clt5 remains in the blur.

**Summary** **7.**
*Finally, by cross-referencing the three sets of results, five clusters stand out, with good statistics on AvStd, density and silhouettes at the same time: clt1, clt2, clt3, clt4, and clt6. Adding the fact that these five clusters are also very large, they can confidently be assimilated to real behaviors—or at least regions of the operating space. Three other clusters are on the edge, with very poor statistics, and with a small cardinality, they can be assimilated to outliers: clt7, clt8, and clt9. Only one cluster, clt5, is more problematic, with mid statistics (but poor density): it might be worthy to be split and redistributed to the neighboring clusters (to be confirmed by the KS test).*


#### 3.2.2. Kolmogorov–Smirnov Test

Let us now have a look at the KS tests, so as to see if we can draw any observations from the data distributions. As for battery 1, the sensor-respective empirical cumulative distribution functions are depicted in Figure 8, computed for each of the twelve dimensions (sensors) of database $D_{2}$, and their means are represented in Figure 9.

From the first figure’s curves, three conclusions can be drawn: (1) the database’s distribution is unique, and no cluster’s is stuck to it; (2) four main groups of curves appear: red–blue–cyan, olive–purple, green–orange–pink, and finally the lone gray; (3) the cluster’s distributions are globally heterogeneous, even though local similarities can be denoted.

From the first point, the high dissimilarities between the cluster’s distributions and that of the database indicate that there are local behaviors and, therefore, the whole database is not homogeneous, and a local clustering approach is well-suited to it.

The second point seems to indicate that the database comprises four real regions within its feature space; this is actually true, with the shutdown, the steady state, and some outlying behaviors. As a result, the conclusions drawn from Section 3.2.1 are emphasized by the empirical CDFs: there seems to be up to four real behaviors. The quantifiers identified five very noticeable clusters, and the four groups of curves tend to say that the order of magnitude is actually that. This information may be used for some form of refinement, such as a new round with a clustering method, but this time comprising no more than five or six clusters (a value a little higher should be preferred to absorb the local outliers).

The third point emphases there is very likely four regions in this database, one per group of curves. What is interesting here is the closeness of some CDFs: for instance, the blue and the cyan are very close, indicating their distributions are very similar. As a result, their respective data seem to belong to the same region and, therefore, the closeness implies the two clusters could be merged without degrading the quality or the representativity of their respective data since they all follow the same distributions. Unfortunately, this is a little too simple, since it is a good thing for the cyan and the blue curves, but not for the orange and the pink ones, which are, respectively, battery 2’s shutdown state and an unwanted event. The closeness of both curves indicates that they are very similar (which is actually true), but without a contextualized interpretation of these results, this would be hard to proceed to a correct anomaly detection analysis using only the empirical CDFs.

Actually, the last point is a consequence of considering only the mean CDFs, even though there actually are twelve dimensions. Indeed, Figure 8 exhibits an important fact: the closeness between the curves mentioned in the previous paragraph are actually true in some dimensions only—the reason why the SOMs created distinct clusters. For instance, the orange and the pink CDFs are very similar in most of the dimensions, except for sensors 7 and 12, where the two curves are very dissimilar. This is also true for the blue and cyan CDFs, which differ from one another in sensors 2, 8, 9, and 12. This means it is very hard to draw any certitude with only these curves, except the number of groups of curves, assumed to be representative of the number of regions of the feature space.

Nevertheless, these conclusions have been drawn from the empirical CDFs only, which are hard to handle—for they are graphical. The true measurement of this section is the KS test, computed for any pair of clusters, and gathered within Table 8. All of the probabilities are expressed in percentages for ease of reading, the matrix is symmetrical, and the score represents how similar two distributions are: the lower the closer.

Interestingly, the KS tests are globally high, meaning the clusters are quite dissimilar; some exceptions are noticeable however. Indeed, clt1, clt4, and clt9 have quite low KS scores among each other: 13.90 for the first pair, 21.30 for the second, and 13.90 for clt1 and clt9; by transitivity, the fact that they are all low means the three cluster’s distributions are close, which is actually true since the three clusters are the three parts of the shutdown state, with, respectively, the core shutdown, the start-up procedure, and the power-off order. Another example of proximity can be denoted with clt2, clt3, clt5, clt6, and clt8; indeed, the pairwise KS tests are quite low, implying again a certain resemblance between their corresponding distributions. Actually, the two first clusters represent the steady state, clt6 is an event that led to a process drift, and the two last one are outliers, which appeared during the transition between the shutdown state and the steady state.

The true question is to know what to do with this information. Indeed, this is not because two distributions are close that they actually represent the same behaviors; however, not merging these clusters would mean ignoring the KS scores, purely and simply. As a consequence, the KS test is worth being considered alongside the previous quantifiers of Section 3.2.1. Indeed, when two clusters have very good results, such as clt1 and clt4, and their pair has a low KS score, they should not be merged, for each cluster can be assumed as representative enough of a local behavior. On the contrary, when two clusters have poor results, but a low KS score, such as clt7 and clt8, a possible merging should really be considered. The last scenario appears when one cluster has good results, but not the other one, and their pair has a low KS score; in this case, both clusters should be merged, for the cluster with the poor results is likely a part of that with the good results.

Therefore, what can we conclude about the CDFs and the KS test? In short, they are neither representative of cluster representativeness nor of their intrinsic quality, but they inform of their proximity, and can partially help point out the issuing clusters, especially the behavioral splits and overlaps, when combined with the quantifiers introduced earlier.

**Summary** **8.**
*The empirical CDFs and the KS tests pointed out four–five distinct data distribution trends among the clusters and, therefore, one may conclude there are as many regions in the feature space and, thus, behaviors. This information emphasizes the conclusions drawn from the quantifiers in the previous section, and also helps us understand that some behaviors have actually been split into several clusters (high similarities between the clusters of some pairs, e.g., clt1 and clt4); nonetheless, we should be careful, since a high similarity does not mean a unique behavior, for it can also be an anomaly, the reason why the quantifiers should be considered so as to leave the already correct clusters alone and avoid a possible loss of a detected anomaly by merging it with a regular behavior.*


#### 3.2.3. Qualitative Validation

Let us confirm (or not) these quantitative, blind, and possibly automated observations by displaying the clusters over time, as in Figure 10, and discuss them.

In the figure, five clusters are distinguishable: orange, green, blue, red, and pink. The first two groups are actually the steady state of the battery, split into two pieces: orange for the beginning of the work week, and green for its other part; there is no reason for this split other than the real regular evolution of the processes. The second two are the shutdown state of the battery, with very similar values, except with the less representative sensor 9, where the data are highly scattered due to the absence of constraints on the process when the plant is off (freewheel). The last cluster represents a real unforeseen event that occurred during the data recording, which forced the system to be shut down prematurely.

Therefore, are our quantitative expectations validated by these qualitative, expert-based ones? The first thing to be noticed is the relevance of the blind clustering, even better than that of battery 1; indeed, here are five clearly distinguishable groups of data, with few overlaps (except clt1 and clt4 perhaps). The two main regions of the operating area, in the feature space, have been identified, although cut in half; a third area has also been extracted, corresponding to a real prohibitive event. Therefore, the blind clustering can qualitatively be validated, although not perfect.

These qualitative observations meet the quantitative ones: clt1, clt2, clt3, clt4, and clt6 are clear compact groups of data, representative of the real (kinds of) behaviors battery 2 can enter in, plus an abnormal event. That being said, what about the problematic cluster clt5, whose AvStd and silhouettes proved it to be of good quality, but not the density? In the figure, it is clear that it comprises only outliers, and can positively not be assimilated to a real behavior on its own. Its data are close together, forming a compact but poorly dense group, the reason for the disagreement between the quantifiers.

All of these statements also confirm the KS tests: there are actually four subregions in the feature space, represented by clt1–clt4, clt2–clt3, clt6, and eventually clt5, so the fact their empirical data distributions create as many trends validates the qualitative observations. Moreover, the mentioned groups of clusters are very close in Figure 10, the reason for their respective pairwise low KS scores. Nonetheless, they should be used as indicators in addition to the previous quantifiers. Indeed, for instance, clt1, clt4, and clt9 had very low pairwise KS scores, meaning they could be merged so as to represent only one major region; this sounds reasonable, since one can assess that they are very close in the graphs. Similarly, clt2 and clt3 may be fused, for they are close and their KS score is low: the resulting cluster would again be more representative of the steady state region. That being said, this is equally true for clt2 and clt6 or clt3 and clt6: they are close in the feature space, have quite low KS scores, and might be merged, but doing so would make us lose important information: clt6 is an anomaly, and should remain alone.

**Summary** **9.**
*The fact that clt1, clt2, clt3, clt4, and clt6 had good results with the quantifiers (AvStd, density, and silhouettes) indicates that they are representative enough of local subregions of the feature space and, therefore, may be better to be left alone. However, clt5 was an issuing cluster, with mid results, and the fact that is it close to clt6 and clt8 according to the KS score would indicate that they all might be worth being fused so as to be more representative of a unique region, which is actually true according to Figure 10, for they are all representative of intermediate values, not the shutdown or the steady state, but more outlier-like, or possible anomalies or events.*


#### 3.2.4. Local Conclusion about Battery 2

This second example is a little more important than the previous one, for it was studied using an automated methodology based on blind quantifiers; of course, we referred to the real clusters and to our knowledge about the systems in order to confirm the results, but the method remained blind and automatized. How did we proceed? We first clustered the system’s database using a self-organizing map, then we characterized the clusters with our quantifiers (AvStd, density, and silhouettes), we confronted their intrinsic properties to decide their qualities (in the sense of the confidence we could put in them), and we used the cluster with the best properties as a reference, for we assumed it was a real behavior of the system (or at least a compact subpart of it).

By doing so, we identified five clusters worthy of attention, for they had good results in their quantifications; they were assimilated to real subregions of the feature space, and the remaining clusters were classified as outliers. Only one cluster was issued, for its quantifiers had mid results, due to its intrinsic nature (a set of outliers). Along with this procedure, we computed the KS scores for every possible pair of clusters, which helped us understand how they were related to each other, and how we could improve the clustering, for instance by fusing the clusters when the KS score was low.

All of these tools proved to be trustworthy when compared to the ground truth with the qualitative analysis of the database. We used much expert-based knowledge to ensure the relevance and trustfulness of our methodology, but it proved to be resilient and representative of the different subregions of the feature space and, therefore, of the behaviors of the system. That knowledge can be used later on for some anomaly tracking purposes, or for local modeling of the system using a multi-model approach.

Finally, let us once more have a short look at the execution times as of Table 9: nothing really new compared to battery 1, the results meet our possible expectations. Again, the AvStd is the fastest method, followed by the density, 1.67 times slower, the KS test, 10.23 times slower, and finally the silhouettes, 86,341 times slower. Once more, and even though the silhouettes might bring complementary, interesting, and relevant information, they are very slow to compute, and can be prohibitive with large databases; moreover, the conclusions we draw from both AvStd and density worked very well side-by-side and were emphasized by the KS tests, and were only comforted by the silhouettes: they might be neglected if there was no doubt about the cluster consistency.

## 4. Conclusions

In this paper, we dealt with the tricky problem of *automatic behavior identification* within an Industry 4.0 context. We proposed using a data-driven and machine learning-based clustering method, namely the efficient self-organizing maps, to gather similar data within compact groups. Moreover, since we worked in a data mining context, we blindly dealt with data about which we could make no expert-based assumptions; we therefore proposed to have recourse to some quantifiers so as to estimate the “quality” of the clusters, i.e., how they were representative of the subregions of a system’s feature space. We used three metrics based on the intrinsic properties of the cluster’s data, namely the average standard deviation, the density, and the silhouette coefficients; we also used the KS test in order to compare the empirical data distributions of the different clusters with each other. We assessed this methodology upon two databases, admittedly connected to one another in some way, recorded during the same period of time, and within the same plant, but whose processes really differed nonetheless.

Self-organizing maps, directly applied to the raw data, provided good results, succeeding in identifying the real behaviors of a system under consideration and even isolating real failures that led to stoppages of the system. Qualitatively speaking, although imperfect, the clustering was very satisfying for both databases, leading to the clear identification of the main regions of the operating spaces, especially the steady state, the shutdown, and some unforeseen, prohibitive events; no overlap appeared, which is very convenient, since the clusters are therefore incomplete, but positively not erroneous.

In order to automatically decide the veracity and representativeness of the clusters, we applied those three quantifiers to each; globally, each metric was able to correctly characterize the main clusters, i.e., to assimilate the clustered, real regions of the operating space as real behaviors, worthy of attention, and of further development, while rejecting the most erroneous. Generally, the simplicity of the AvStd led to some limitations, but was compensated by the density, which was often more accurate and more relevant, for it distinctly rejected the issuing clusters, but not the largest and most representative ones. In case of doubt, the silhouette coefficients (perhaps the most representative of the three quantifiers) allowed to finally decide; for there was one coefficient per datum, it was also the slowest—by tens of thousands of times—which might be prohibitive in many contexts.

Once the clusters were characterized, we analyzed and compared their data distributions, and we computed the KS test upon every possible pair of clusters so as to point out the similar ones, which might be worth being merged. The test provided us with highly valuable information, but without context, merging two clusters may lead to a loss of information; for instance, this may be the case with an anomaly whose data would be similar to that of regular behavior, and with the regular behavior in question. An anomaly is generally more isolated and lasts a shorter time; therefore, such a merging would make us lose the information that one of the clusters is actually an anomaly. To avoid that, we showed that we could consider an upstream characterization of the clusters. Indeed, the KS test does not aim to characterize a set of data, it is only useful to compare different sets: it does not consider the intrinsic properties of the datasets, which might lead to some misleadings. As a consequence, we showed that the KS test should be taken into account for a possible merging of two clusters with similar data distributions only when at least one them had poor results with the quantifiers, for it might be a subpart of the behavior represented by the other cluster, which would have been separated from it by the clustering. A short summary of these observations is depicted in Figure 11.

These metrics, especially the density, were able to characterize both good quality and bad quality clusters: a large, compact and dense cluster has good results, but a cluster that spans across two (or more) behaviors have poor results, indicating a problem.

These results are encouraging, as they show that automatic behavior identification can be applied to real Industry 4.0 data, with great accuracy and relevance. They rely on a simple and efficient two-step methodology: (1) machine learning-based clustering used to identify and isolate the different regions of the feature space; and (2) some quantifications, such as AvStd and density, used to assess the quality of the clusters. It is easy to use and conceptualize: positively assuming nothing about the data but very general information, it can be used to automatically make sense of the data, as blind users. It is an unsupervised, valuable manner to clear the way for higher levels of treatment.

This work is only one step toward Industry 4.0 and the cognitive plant: it paves the way for a higher level and more complex understanding of the industrial systems. There is still much to do though; the next steps are multiple, e.g., improving the clustering, improving the quantifiers, extending to more cases to assess the universality of the approach, and modeling sensu stricto of the systems, based on the so-built clusters, following a multi-model approach, for instance.

### Future Work

This last point will probably constitute the future steps of our work, based on this methodology. Indeed, we strongly believe that a multi-model approach is well suited to industrial, dynamic processes, as investigated in [21]. The idea was to split a system within its temporal space and then apply local models (MLP, RBF, etc.) upon each of these local, short-time intervals. We believe that a feature-based approach is better suited in such a context, for it allows modeling the system according to its intrinsic behaviors rather than based on its only past. The work presented in this present article serves as a blind preprocessing for the modeling: once the database’s feature space is split into subregions, assumed to be representative of the real behaviors of the system, and the quality of the clusters are checked using the previous quantifiers, it is possible to refine the clustering, either by splitting some weird and issuing clusters, or by merging some of them so as to obtain larger clusters, which would cover a wider region within the feature space. All of these elements serve for the next great step: the local modeling of each behavior (represented by a unique region) and then the fusion of all the local behaviors, so as to model the whole system, as a mixture of its different possible and observed behaviors.

## Figures and Tables

**Figure 1 sensors-22-02939-f001:**
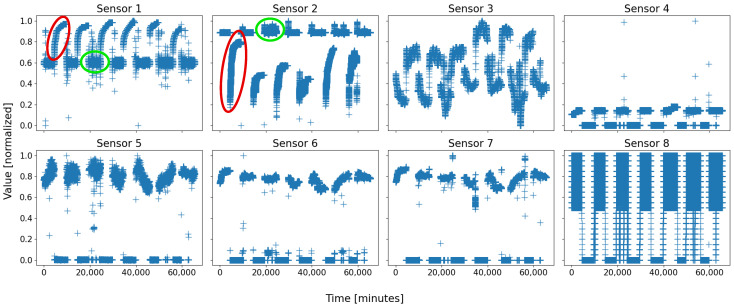
Battery 1 ($D_{1}$) sensors over time. Two main motifs stand out, circled in red and green: the green coarsely corresponds to the steady state of the process, whereas the red is more the state of the sensors when the plant is shut down. A pair of red and green motifs forms a regular work week.

**Figure 2 sensors-22-02939-f002:**
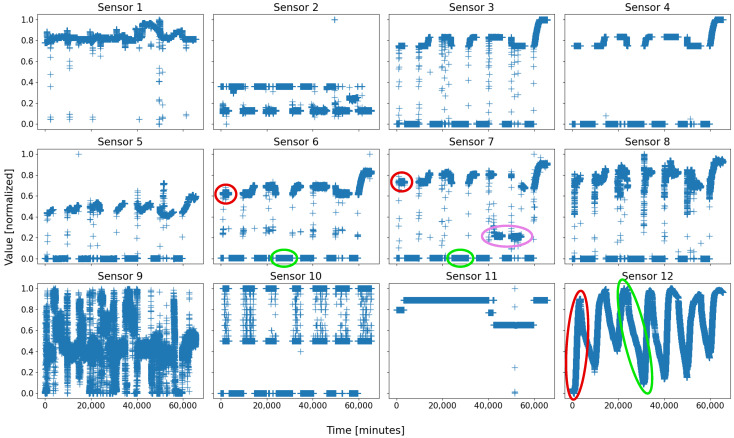
Battery 2 ($D_{2}$) sensors over time. Similar to battery 1 (Figure 1), two main motifs stand out, circled in red and green, and a pair of both represents, anew, a regular work week, with the steady state in green and the offline state in red, respectively. A third group of data, circled in pink, is also noticeable: it represents an irregular event that occurred during the data recording.

**Figure 3 sensors-22-02939-f003:**
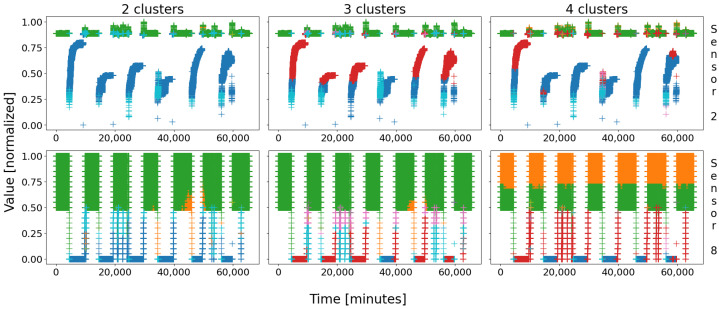
Battery 1 ($D_{1}$) clustering comparison (3×3 SOM). When using a SOM with $D_{1}$, one among three sets of clusters appears; each column is one of these configurations, and the rows are two sensors of the database given as examples, sensor 2 (**top**) and sensor 8 (**bottom**).

**Figure 4 sensors-22-02939-f004:**
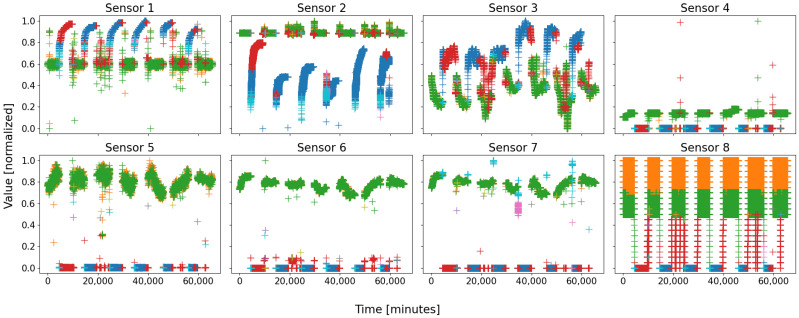
Battery 1’s eight clustered sensors of the rightmost column of Figure 3.

**Figure 5 sensors-22-02939-f005:**
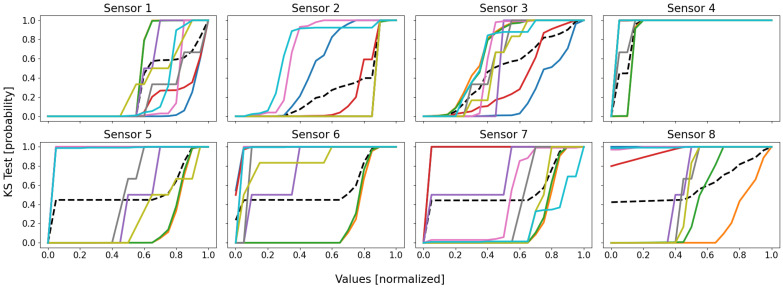
Battery 1 probability distributions. The solid colored lines are the empirical distributions of each cluster, and the black dashed line is that of the database $D_{1}$. The abscissa shows the real normalized values of the corresponding sensor, and the ordinates, their probabilities of appearance.

**Figure 6 sensors-22-02939-f006:**
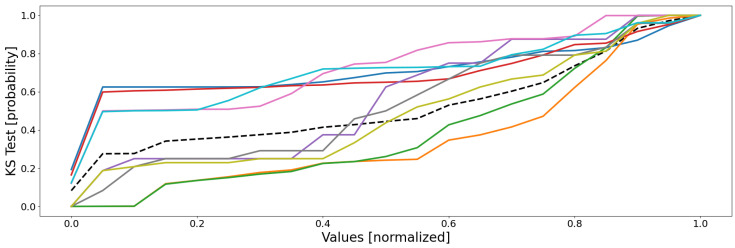
Battery 1 mean probability distributions computed along each dimension (sensor). The solid color lines are those of the clusters, and the dashed black line is that of the database. Normalized sensor values in abscissa and empirical probability of appearance in the ordinate.

**Figure 7 sensors-22-02939-f007:**
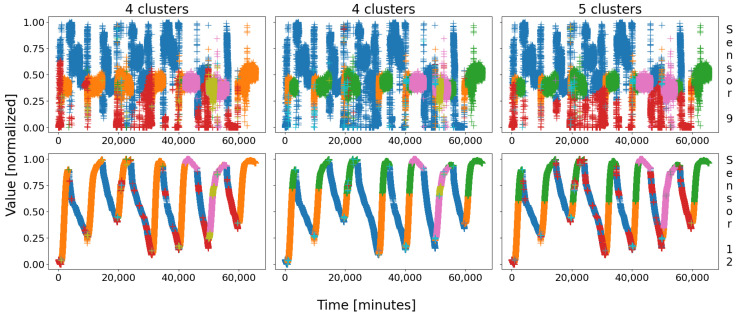
Battery 2 ($D_{2}$) clustering comparison (3×3 SOM). The SOMs end up with three possible configurations, with four or five main clusters; these configurations are the columns, whilst the rows are two of the sensors of the database, sensor 9 (**top**) and sensor 12 (**bottom**).

**Figure 8 sensors-22-02939-f008:**
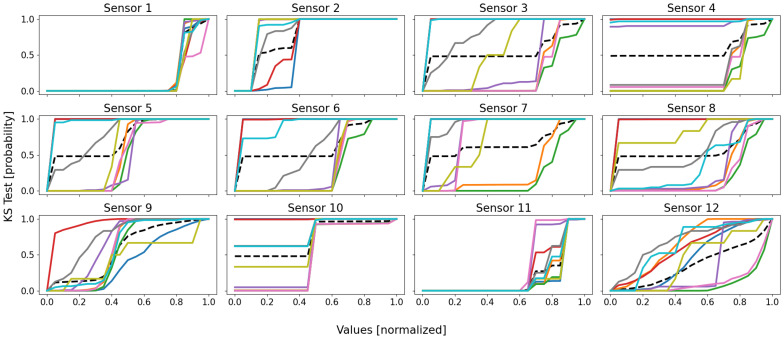
Battery 2 probability distributions. The solid colored lines are the empirical distributions of each cluster, and the black dashed line is that of the database $D_{2}$. The real normalized values of the corresponding sensor are in abscissa, and their respective probabilities of appearance are in ordinate.

**Figure 9 sensors-22-02939-f009:**
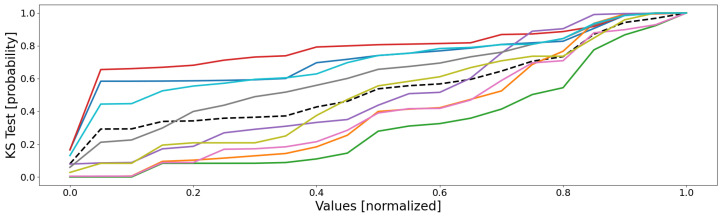
Battery 2 mean probability distributions computed along each dimension (sensor). The solid color lines are those of the clusters, and the dashed black line is that of the database. Normalized sensor values in abscissa and empirical probability of appearance in ordinate.

**Figure 10 sensors-22-02939-f010:**
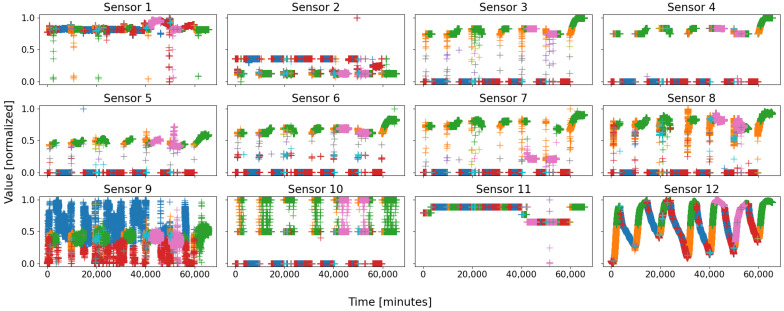
Battery 2’s twelve clustered sensors of the rightmost column of Figure 7.

**Figure 11 sensors-22-02939-f011:**
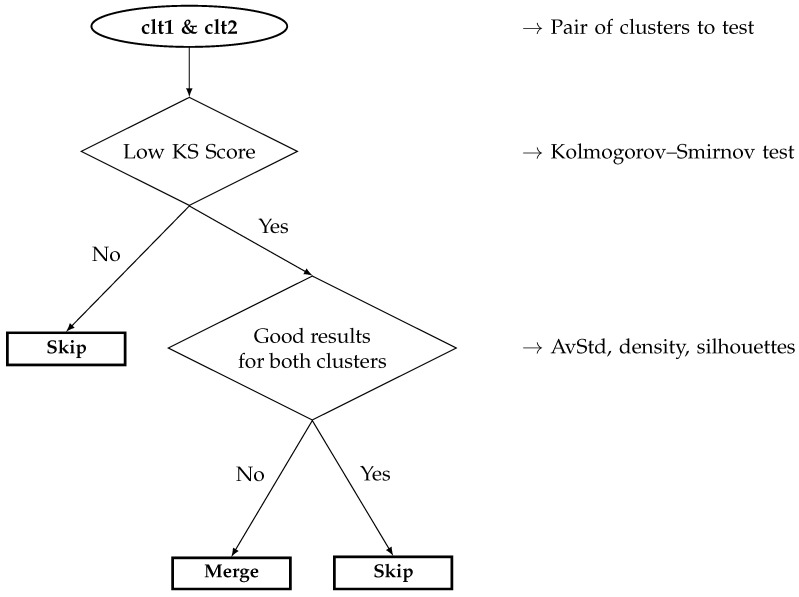
Summary of the fusion of the information brought by the quantifiers and the KS scores.

**Table 1 sensors-22-02939-t001:** Recapitulation of the quantifiers.

Name	Acronym	Description	Interpretation
Kolmogorov–Smirnov test	KS test	Statistical comparison of two datasets based on the empirical data distributions.	The lower the score, the higher the resemblance between the two datasets.
Average Standard deviation	AvStd	Mean of the standard deviations computed along all the dimensions.	Compactness of the cluster, i.e., how the data are centered around their means.
Hyper-Density	Density	Smallest ND hypersphere covering all of the cluster’s data.	Density of the cluster, i.e., how full and scattered the hyper-volume is.
Silhouette Coefficients	SCs	Normalized differences between the intra-cluster distances of the data and the inter-cluster distances.	The higher, the nearer to the data of the same cluster and the father from the data of the other clusters, and *vice-versa*.

**Table 2 sensors-22-02939-t002:** AvStd and density of the two databases.

Database		Average Standard Deviation			Density
Tag	Card	Min	Mean	Max	
$D_{1}$	65,505	0.071	0.280	0.405	44,401.066
$D_{2}$	63,715	0.048	0.267	0.414	39,935.226

**Table 3 sensors-22-02939-t003:** Cluster color scheme.

**Clusters**	clt1	clt2	clt3	clt4	clt5	clt6	clt7	clt8	clt9
**Colors**									
	Blue	Orange	Green	Red	Purple	Pink	Gray	Olive	Cyan

**Table 4 sensors-22-02939-t004:** Battery 1 pairwise KS test of the clusters. All probabilities expressed in percentages (×100). See Table 3 for the details about the color scheme.

	clt1	clt2	clt3	clt4	clt5	clt6	clt7	clt8	clt9
**clt1**	0.00	62.50	62.40	6.40	43.80	16.80	54.20	43.80	12.90
**clt2**	62.50	0.00	12.00	60.40	45.90	57.10	37.60	27.40	49.90
**clt3**	62.40	12.00	0.00	60.40	38.00	51.00	27.60	21.30	49.90
**clt4**	6.40	60.40	60.40	0.00	41.10	18.90	51.50	41.10	10.90
**clt5**	43.80	45.90	38.00	41.10	0.00	37.00	12.50	20.80	41.90
**clt6**	16.80	57.10	51.00	18.90	37.00	0.00	41.70	44.50	12.80
**clt7**	54.20	37.60	27.60	51.50	12.50	41.70	0.00	12.50	42.80
**clt8**	43.80	27.40	21.30	41.10	20.80	44.50	12.50	0.00	46.90
**clt9**	12.90	49.90	49.90	10.90	41.90	12.80	42.80	46.90	0.00

**Table 5 sensors-22-02939-t005:** Battery 1 quantification of the clustering: AvStd, density, and silhouettes. For the first two tags, “Abs” represents the absolute value of quantifier γ, “$÷\;\gamma_{D}$” is this value divided by that of database $D_{1}$, and “$÷\;\gamma_{\max}$”, divided by the maximal value of γ among the nine clusters. For the silhouettes, there are as many coefficients as data in a cluster: “Min”, “Mean”, and “Max” are, respectively, the minimum, mean, and maximum values among all the silhouettes of a cluster. See Table 3 for the details about the color scheme.

Clusters		AvStd $\bar{\sigma }$			Density ρ			Silhouettes SC		
**Tag**	**Card**	**Abs $\left(\times 10^{-2}\right)$**	$÷\;{\bar{\sigma }}_{D}$	$÷\;{\bar{\sigma }}_{\max}$	**Abs**	$÷\;\rho_{D}$ $\left(\times 10^{-2}\right)$	$÷\;\rho_{\max}$ $\left(\times 10^{-2}\right)$	**Min**	**Mean**	**Max**
**clt1**	21,178	3.477	0.124	0.401	26,066	58.7	89.3	−0.060	0.496	0.633
**clt2**	21,210	4.784	0.171	0.553	29,186	65.7	100	−0.165	0.314	0.541
**clt3**	15,034	4.662	0.167	0.539	14,365	32.4	49.2	0.028	0.374	0.556
**clt4**	7747	6.715	0.240	0.777	5548	12.5	19.0	−0.289	0.291	0.439
**clt5**	2	7.250	0.259	0.838	5.286	0.0119	0.0181	−0.588	−0.211	0.166
**clt6**	102	3.487	0.125	0.403	129.2	0.291	0.443	−0.202	0.666	0.781
**clt7**	3	5.606	0.200	0.648	11.94	0.0269	0.0409	0.192	0.277	0.381
**clt8**	6	8.648	0.309	1.000	10.12	0.0228	0.0347	−0.257	−0.012	0.192
**clt9**	223	7.900	0.282	0.913	147.8	0.333	0.506	−0.616	0.085	0.442

**Table 6 sensors-22-02939-t006:** Battery 1 execution times of the quantifiers (all values in milliseconds, ms).

AvStd			Density			Silhouettes		KS Test	
Abs	$÷\;{\bar{\sigma }}_{D}$	$÷\;{\bar{\sigma }}_{\max}$	Abs	$÷\;\rho_{D}$	$÷\;\rho_{\max}$	Abs	Stats	CDFs	Scores
3.796	3.885	0.010	6.787	7.452	0.012	$4.001\times 10^{5}$	0.330	98.877	0.868
7.691 ms			14.251 ms			$4.001\times 10^{5}$ ms		99.745 ms	

**Table 7 sensors-22-02939-t007:** Battery 2 quantification of the clustering: AvStd, density, and SCs. “Abs” is the absolute value of the quantifier γ, whilst “$÷\;\gamma_{D}$” is this value divided by that of the database, and “$÷\;\gamma_{\max}$” by its maximum. “Min”, “Mean”, and “Max” are the minimum, mean, and maximum of the silhouettes. See Table 3 for the details about the color scheme.

Clusters		AvStd $\bar{\sigma }$			Density ρ			Silhouettes SC		
**Tag**	**Card**	**Abs $\left(\times 10^{-2}\right)$**	$÷\;{\bar{\sigma }}_{D}$	$÷\;{\bar{\sigma }}_{\max}$	**Abs**	$÷\;\rho_{D}$ $\left(\times 10^{-2}\right)$	$÷\;\rho_{\max}$ $\left(\times 10^{-2}\right)$	**Min**	**Mean**	**Max**
**clt1**	21,514	4.41	0.164	0.287	18,216	45.6	100	−0.134	0.489	0.653
**clt2**	7199	7.05	0.261	0.458	6873	17.2	37.7	−0.190	0.427	0.616
**clt3**	18,369	6.93	0.258	0.453	17,393	43.6	95.5	−0.253	0.498	0.665
**clt4**	9091	5.68	0.208	0.365	7737	19.4	42.5	−0.398	0.439	0.587
**clt5**	104	8.42	0.316	0.554	88.52	0.222	0.486	−0.368	0.640	0.781
**clt6**	7345	7.01	0.262	0.460	6909	17.3	37.9	−0.725	0.490	0.648
**clt7**	24	15.2	0.570	1.000	21.42	0.0536	0.118	−0.362	0.128	0.388
**clt8**	6	11.6	0.434	0.760	8.387	0.0210	0.0461	−0.182	0.153	0.307
**clt9**	63	10.7	0.400	0.701	48.55	0.122	0.267	−0.273	0.363	0.530

**Table 8 sensors-22-02939-t008:** Battery 2 pairwise KS test of the clusters. All probabilities expressed in percentages (×100). See Table 3 for the details about the color scheme.

	clt1	clt2	clt3	clt4	clt5	clt6	clt7	clt8	clt9
**clt1**	0.00	58.00	58.70	13.90	49.80	57.90	37.20	50.10	13.90
**clt2**	58.00	0.00	22.20	65.50	23.10	9.50	37.50	21.80	46.60
**clt3**	58.70	22.20	0.00	68.20	38.60	19.30	45.50	32.70	55.20
**clt4**	13.90	65.50	68.20	0.00	57.20	65.60	44.30	57.70	21.30
**clt5**	49.80	23.10	38.60	57.20	0.00	19.50	25.10	16.80	36.80
**clt6**	57.90	9.50	19.30	65.60	19.50	0.00	34.40	19.70	46.60
**clt7**	37.20	37.50	45.50	44.30	25.10	34.40	0.00	28.10	23.30
**clt8**	50.10	21.80	32.70	57.70	16.80	19.70	28.10	0.00	38.70
**clt9**	13.90	46.60	55.20	21.30	36.80	46.60	23.30	38.70	0.00

**Table 9 sensors-22-02939-t009:** Battery 2 execution times of the quantifiers (all values in milliseconds, ms).

AvStd			Density			Silhouettes		KS Test	
Abs	$÷\;{\bar{\sigma }}_{D}$	$÷\;{\bar{\sigma }}_{\max}$	Abs	$÷\;\rho_{D}$	$÷\;\rho_{\max}$	Abs	Stats	CDFs	Scores
4.788	5.123	0.010	8.007	8.324	0.013	$4.134\times 10^{5}$	0.323	100.786	0.668
9.921 ms			16.343 ms			$4.134\times 10^{5}$ ms		101.454 ms	

## Data Availability

Private industrial data; not publicly available.

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
