# Peer review of "Clustering at the Disposal of Industry 4.0: Automatic Extraction of Plant Behaviors [Author-notes fn1-sensors-22-02939]"

_sensors, 2022, doi:10.3390/s22082939_

Round 1
Reviewer 1 Report
The authors have applied Self-Organizing Maps, which is a Machine Learning methodology, to cluster two datasets from a chemistry plant and have studied three metrics to quantify the clusters and characterize the behavior of the plant. In my opinion, the paper is very comprehensive and scientifically sound. Furthermore, it makes a novel contribution to the Industry 4.0 literature.
Reviewer 2 Report
The paper is of interest but the current form is not suitable for publication.
Introduction: It is too long and diverse. I would suggest a significant revision there.
In terms of new results, I would suggest using a statistical test such as the Hassani and Silva Test which is a non-parametric test and works well for any type of comparison, '' A Kolmogorov-Smirnov Based Test for Comparing the Predictive Accuracy of Two Sets of Forecasts''.
The clustering results are very promising but would be good to mention the challenges as well.
Reviewer 3 Report
The list of references should be increased. If it is research paper, the literature survey is required. I would ask if the colors have the same weights? The paper requires a ceratin integration, as it is visible that it is written by individuals,
please, expand the future research section,
further comments are included in the attached file

Round 2
Reviewer 3 Report
Authors tried to radically improve their paper. Now it contains many details, explanations, experiments, so it looks like a research report. May be authors should take role of reader and answer quuestion what is important from the point of view of readers. Interpretation of measures? Experiments results?
May be paper can be slightly reduced, but it is only my suggestion.
However, beyond that , paper is valuable and should be published.
